# Compound C Inhibits B16-F1 Tumor Growth in a Syngeneic Mouse Model via the Blockage of Cell Cycle Progression and Angiogenesis

**DOI:** 10.3390/cancers11060823

**Published:** 2019-06-13

**Authors:** Yun Taek Lee, So Hyun Lim, Boram Lee, Insug Kang, Eui-Ju Yeo

**Affiliations:** 1Department of Medicine, Gachon University School of Medicine, Incheon 21999, Korea; Soboq0208@gmail.com; 2Department of Biochemistry, Gachon University School of Medicine, Incheon 21999, Korea; Dlathgus1@naver.com (S.H.L.); Boram.lee.anne@gmail.com (B.L.); 3Department of Biochemistry and Molecular Biology, Medical Research Center for Bioreaction to Reactive Oxygen Species and Biomedical Science Institute, Graduate School, Kyung Hee University School of Medicine, Seoul 02447, Korea

**Keywords:** compound C, cell cycle arrest, angiogenesis, VEGFR, B16-F1, HUVECs, C57BL/6

## Abstract

We recently observed that Compound C (CompC), a reversible inhibitor of AMP-activated protein kinase, reduced the cell viability of B16-F1 melanoma cells. To establish its molecular mechanism(s) of action, the cell cycle was examined by flow cytometry and the expression of cell cycle regulatory proteins and angiogenesis-related proteins were examined by western blot analysis. In addition, its effect on tumor growth was investigated using C57BL/6 syngeneic mice bearing B16-F1 xenografts. We found that CompC induced G2/M cell cycle arrest, which was associated with reduced levels of cell cycle regulatory proteins, such as phosphorylated pRB, cyclin-dependent protein kinases (Cdks), cyclins, and phosphorylated P-Ser^10^-histone H3, and increased levels of Cdk inhibitors, such as p21 and p53. We also found that CompC inhibits proliferation, migration, and tube formation of human umbilical vascular endothelial cells via the inhibition of vascular endothelial growth factor receptor-induced signaling pathways. As expected, CompC significantly reduced the tumor size of B16-F1 xenografts in the syngeneic mouse model. Inhibition of tumor growth may be attributed to reduced cell proliferation via cell cycle inhibition and in part to decreased angiogenesis in CompC-treated mice. These findings suggest the potential use of CompC against melanoma development and progression.

## 1. Introduction

Recently Compound C (6-[4-(2-piperidin-1-yl-ethoxy)-phenyl]-3-pyridin-4-yl-pyrazolo[1,5-*a*]-pyrimidine, CompC), a selective and reversible inhibitor of AMP-activated protein kinase (AMPK), was shown to reduce the cell viability of human diploid fibroblasts (HDFs) [1]. CompC-induced growth arrest was correlated with a decrease in the cell cycle regulatory proteins, such as proliferating cell nuclear antigen (PCNA), phosphorylated pRb (P-pRb), cyclin-dependent protein kinases (Cdk2 and Cdk4), cyclins (D and E) via reduction of platelet-derived growth factor (PDGF)-induced tyrosine phosphorylation of PDGF receptor β (PDGFRβ) and the activity of downstream signaling molecules, such as p85, phosphatidylinositol 3-kinase (PI3K), phospholipase C-γ1 (PLCγ1), and extracellular signal-regulated kinase 1/2 (ERK1/2) [1]. Although AMPK has been associated with antiproliferative actions in a range of cell systems [2,3], the biological effects of CompC seem to be AMPK-independent. The inhibitory effect of CompC on PDGFR phosphorylation and downstream signaling events was independent of AMPK activity in HDFs [1]. AMPK-independent antiproliferation in glioblastoma cells by CompC was also suggested by Liu et al. [4]. Because the selectivity of CompC for AMPK remains uncertain, it was postulated that CompC may induce another pathway for cell cycle arrest and apoptosis [5]. It is necessary to clarify whether CompC exerts an inhibitory effect on the proliferation of other cancer cells.

Vascular endothelial growth factors (VEGFs) are important regulators of vascular development during angiogenesis [6]. VEGF includes five isoforms that bind three types of VEGF receptors (VEGFR): VEGFR1, VEGFR2 and VEGFR3. VEGFR has structural similarity to PDGFR. Both PDGFR and VEGFR have seven immunoglobulin-like extracellular domains [7]. Similar to PDGF-PDGFR binding, the dimerization and activation of VEGFR can be induced by the binding of VEGF [8]. Therefore, the expected responses in VEGFR signaling after CompC treatment could be similar to those in PDGFR signaling. For tumor angiogenesis, the tyrosine phosphorylation of VEGFR is crucial, and the deficiency of these sites causes defects in tumor vascularization and tumor growth [9]. In addition, the PDGF/PDGFR family stimulates the proliferation of mesenchymal origin cells, such as fibroblasts, and these cells have supporting roles in angiogenesis [10]. CompC-induced inhibition of PDGFR signaling and proliferation in fibroblasts might also be responsible for the reduction of tumor vascularization and growth.

In this study, we examined the effect of CompC on the viability of B16-F1 mouse melanoma cells. The effects of CompC on the expression and phosphorylation of cell signaling proteins and cell cycle regulatory proteins were examined in B16-F1 cells. Because human umbilical vein endothelial cells (HUVECs) have specific signaling pathways for VEGF [11] and provide a critical in vitro model for the regulatory mechanisms of angiogenesis or neovascularization in tumors [12,13], the effects of CompC on cell viability, tube formation, and cell migration were examined in HUVECs. Furthermore, to clarify whether CompC affects tumor growth and angiogenesis in vivo, the effect of CompC on tumor growth and intratumor neovascularization was examined in B16-F1 tumor-bearing 1. C57BL/6 syngeneic mice.

## 2. Results

### 2.1. CompC Reduces the Viability of B16-F1 Melanoma Cells via G2/M Cell Cycle Arrest

The effect of CompC on cell viability was determined in various melanoma cells, including B16-F1, B16-BL6 and K-1735 murine melanoma cells and G361, DX3, A375P (low metastatic) and A375SM (highly metastatic) human melanoma cells, and normal human epidermal melanocytes from adults (NHEMa). Cells were treated with vehicle (DMSO) or various concentrations (1–20 μM) of CompC for 2 days in the culture media. The cell viability was measured by the 3-(4,5-Dimethylthiazol-2-yl)-2,5-diphenyltetrazolium bromide (MTT) assay. The viability of various melanoma cells was similarly decreased by CompC treatment in a dose-dependent manner (Figure 1A). As shown in Figure 1A, the cytotoxic effect of CompC was much greater in murine and human melanoma cells compared to that in normal human epidermal melanocytes. The IC_50_ values of CompC for inhibition of each cells at 48 h are determined as 3.3 (B16-F1), 5.2 (B16-BL6), 5.2 (K-1735), 8.8 (G361), 6.5 (DX3), 4.8 (A375P), 4.7 (A375SM), and 16.3 μM (NHEMa). Because CompC is most effective for B16-F1 cells compared to other melanoma cells in the present study, we chose B16-F1 cells for further studies of the effect of CompC on melanoma cells. Morphological changes of B16-F1 cells were also examined after incubation with 10 μM CompC for 2–4 days. Changes in cell shape and detachment of cells were clearly visible after treatment with 10 μM CompC for 2 days (Figure 1B).

To determine whether CompC inhibits B16-F1 cell proliferation by cell cycle arrest, the DNA contents were measured by flow cytometry after staining the nuclei with propidium iodide (PI). Serum-starved cells were stimulated with serum in the presence of vehicle or 10 μM CompC for 8–24 h. The data showed that CompC increased cells in the G2/M fraction from 24.2% to 44.3%, while it reduced cells in the G1 phase from 54.0% to 26.6% at 24 h (Figure 1C). CompC did not alter the number of cells in the S phase. Flow cytometry also revealed that there was only a slight increase in apoptotic cell death as judged by the altered sub-G1 fraction in CompC-treated cells: from 2.5% to 9.4% at 24 h. These results suggest that CompC inhibits cell proliferation mainly by inducing G2/M cell cycle arrest without cell cycle progression to G1 phase through mitosis in B16-F1 cells.

### 2.2. CompC Reduces the Levels of Cdks and Cyclins and Increases the Levels of Cdk Inhibitors in B16-F1 Melanoma Cells

To clarify how CompC induces G2/M cell cycle arrest in B16-F1 cells, the effect of 10 μM CompC on the expression and phosphorylation status of various cell cycle regulatory proteins was examined by western blot analysis. CompC reduced the levels of phosphorylated pRb on Ser^795^ (P-Ser^795^-pRb), total pRb, Cdk4, cyclin D1, Cdk2 and cyclin A at 16–24 h (Figure 2A and Appendix A), which might cause delayed progression through the G1, G1/S transition, and S phase of the cell cycle. CompC treatment also reduced FBS-induced phosphorylated Cdc2 (P-Tyr^15^-Cdc2), Cdc2, cyclin B1, and phosphorylated histone H3 on Ser^10^ (P-H3) at 16 and 24 h (Figure 2A and Appendix A). These results suggested that CompC exerts its inhibitory effect on mitosis via the reduction of Cdc2 kinase activity and subsequent phosphorylation of histone H3.

The Cdk activities can be inhibited by binding Cdk inhibitors, such as p21, p27, and p16, to Cdk-cyclin complexes. Therefore, the levels of these Cdk inhibitors were also examined by western blot analysis. Although the effects of CompC on Cdk inhibitors were different from each other at the early time points (8 and 16 h), the levels of these Cdk inhibitors were obviously increased at 24 h after CompC treatment (Figure 2B and Appendix A). CompC treatment also increased the levels of phosphorylated p53 and intact p53 at 8–16 h, which might play a role in the delayed increase in p21 expression at 24 h. These results suggest that the inhibition of Cdk activity appears to be due in part to increased levels of Cdk inhibitors.To determine whether AMPK inactivation is a critical factor in CompC-induced G2/M cell cycle arrest in B16-F1 melanoma cells, these cells were transfected with AMPKα1/2 siRNA for 48 h, followed by treatment with 10 μM CompC for 16 h. Compared with those of mock siRNA-transfected cells, knockdown of AMPKα1/2 alone enhanced the levels of P-Cdc2 and P-H3, indicating an anti-proliferative effect of AMPK in B16-F1 cells. CompC treatment further reduced the levels of P-Cdc2 and P-H3 in AMPK-downregulated cells (Figure 2C and Appendix A). This result suggests that CompC plays a critical role in G2/M cell cycle arrest in an AMPK-independent manner in B16-F1 cells.

### 2.3. CompC Enhances the Level of Phosphorylated Akt and ERK1/2 via ROS Production in B16-F1

To clarify how CompC affects cell cycle regulatory proteins and Cdk inhibitors in B16-F1 cells, the effect of CompC on upstream signaling proteins, such as Akt and ERK1/2, was examined by western blot analysis. The level of phosphorylated Akt was increased by serum treatment, and interestingly, its level was slightly higher in CompC-treated cells (Figure 3A and Appendix A). In addition, CompC increased the levels of phosphorylated/activated ERK1/2 in both SFM- and FBS-treated cells (Figure 3A and Appendix A). The levels of other proteins, JNK, P-JNK, p38, P-p38, PI3K, P-PI3K, PLCγ1, and P-PLCγ1, were not altered significantly by either serum or CompC treatment. These results suggest that the effect of CompC may be mediated in part through the activation of Akt and ERK1/2 in B16-F1 cells.

Because reactive oxygen species (ROS) were shown to activate Akt and MAPK pathways [14], CompC-induced P-Akt and P-ERK1/2 levels were examined in the absence or presence of 5 mM N-acetyl cysteine (NAC) pretreatment for 1 h. Because CompC-induced P-Akt and P-ERK1/2 levels at 10 and 60 min were decreased by NAC pretreatment (Figure 3B and Appendix A), ROS production might play a role in CompC-induced activation of these proteins. Furthermore, to understand the functional role of CompC-induced ROS production in G2/M cell cycle arrest, cells were pretreated with 5 mM NAC for 1 h, followed by treatment with 10 μM CompC for 16 and 24 h, and the cell cycle was examined by flow cytometry. Treatment with CompC alone resulted in a significant increase in G2/M-arrested cells, and the gated percentage of G2/M-arrested cells was significantly reduced by pretreatment with NAC at 16 h and 24 h (*p* < 0.001) (Figure 3C). In contrast to a reduction of the % gated G2/M, CompC increased significantly (*p* < 0.001) the % gated fractions of sub-G1 and G1 at 16 h and the sub-G1 fraction at 24 h. These data suggest that CompC-induced cell cycle arrest at G2/M might be caused in part through ROS production. CompC-induced ROS production was confirmed at 1 h and 6 h in both SFM- and FBS-treated cells (Figure 3D).

### 2.4. CompC Inhibits HUVEC Cell Viability, Tube Formation, and Cell Migration via the Inhibition of VEGF-Induced Signal Transduction

Previously, the inhibitory effect of CompC on PDGFR signaling was shown in HDFs [1]. Because there is structural similarity between PDGFR and VEGFR [15], it is postulated that the function and downstream signaling of VEGFR would also be reduced by CompC. The effect of CompC on cell viability was first studied in HUVECs, which have abundant VEGFRs on their cell surface membranes [16]. These cells were treated with vehicle or CompC (1–20 μM) for 4 days in the culture medium and the cell viability was examined by MTT assay. CompC reduced the viability of HUVECs in a dose-dependent manner (Figure 4A).

The effect of CompC on the vascularization of HUVECs was then examined by the tube forming assay. HUVECs seeded on Matrigel-coated wells were cultured in endothelial growth media kit 2 (EGM-2) with vehicle or 1–10 μM CompC for 18 h and then stained with Diff-Quik. The level of tube formation and its continuity was reduced by CompC in a dose-dependent manner (Figure 4B). This study also examined the effect of CompC on the migration of HUVECs. The HUVECs to 70–80% confluence were serum-starved by incubation with endothelial cell basal medium (EBM) for 24 h, and some areas were denuded. The cells were then incubated in EGM-2 medium containing vehicle or CompC. Cell migration was significantly increased by EGM-2, but cotreatment with EGM-2 and CompC greatly inhibited cell migration in a dose- and time-dependent manner (Figure 4C–E).

Because HUVECs possess VEGFR2, a major receptor for VEGF-induced signaling, and respond well to human VEGF, the effect of CompC on VEGF-induced signaling was also investigated. Subconfluent HUVECs were pretreated with vehicle or 10 μM CompC for 1 h, treated with 50 ng/mL hVEGF for the indicated times, and then the levels of total and phosphorylated signaling proteins were examined by western blot analysis. The VEGF-induced tyrosine phosphorylation of hVEGFR2 and downregulation of total hVEGFR2 were completely blocked by CompC pretreatment. Although CompC alone weakly increased the basal level of phosphorylated ERK1/2 in HUVECs, as shown in B16-F1, VEGF-induced phosphorylation of p38, ERK1/2, PLCγ1, PKCθ and PKCμ was completely blocked by CompC treatment (Figure 4F and Appendix A). These results suggest that CompC inhibits VEGF-induced signal transduction in HUVECs. The inhibitory effects of CompC on HUVEC viability, tube formation and cell migration and VEGF-induced signal transduction indicate that CompC may have an inhibitory effect on angiogenesis.

### 2.5. CompC Inhibits B16-F1 Tumor Growth and Angiogenesis in C57BL/6 Syngeneic Mice

Because CompC inhibited cell division and angiogenesis in vitro, it was postulated that CompC would inhibit in vivo B16-F1 tumor growth and angiogenesis in C57BL/6 syngeneic mice. B16-F1 cells were subcutaneously injected into C57BL/6 mice and CompC was intraperitoneally injected into the tumor-bearing C57BL/6 mice once a day for 2 weeks. During the experimental periods, the body weight and B16-F1 tumor volume were measured every day. The plot of body weight did not show statistically significant difference between vehicle- and CompC-treated groups (Figure 5A). However, there was a significant difference between treatment groups (χ^2^ = 14.5, df = 1, *p* < 0.001, Kruskal-Wallis nonparametric test). Post hoc comparisons showed statistically significant differences between vehicle and CompC treatment groups on a certain day (Mann-Whitney U = 0.0, *p* < 0.05, Mann-Whitney U tests). Significant difference between vehicle and CompC injection groups at each day is marked with an asterisk (*p* < 0.05, Mann-Whitney U test). The in vivo growth rate of B16-F1 tumors in CompC-injected mice was lower than that of vehicle-treated mice (Figure 5B). The average tumor size in the CompC-treated group was reduced by more than 50% compared with that in the vehicle-treated group.

To clarify the effect of CompC on in vivo angiogenesis during tumorigenesis of B16-F1, intratumor vascularization was examined. The paraffin sections of the excised B16-F1 tumor were H&E-stained and photographed with a light microscope (Olympus, 40–400×). As shown in Figure 5C, vascular endothelial cell layer and blood cells in the stained specimen were clearly observed in the vehicle-treated group but not in the CompC-treated groups. The effect of CompC on in vivo angiogenesis was also confirmed by immunofluorescence staining with fluorescein isothiocyanate (FITC)-labeled antibodies against panendothelial cell antigen MECA32 in frozen sections of the excised B16-F1 tumor. Confocal microscopic observation revealed that CompC treatment greatly reduced the blue fluorescent FITC signal, which indicated the presence of vascular endothelial cells (Figure 5D). Negative controls were prepared without anti-MECA32 antibodies in the presence of FITC-labeled secondary antibodies and 4’,6-diamidino-2-phenylindole (DAPI) stain (Figure 5D). These data suggested that CompC could inhibit in vivo angiogenesis during tumorigenesis of B16-F1.

## 3. Discussion

In a previous study, the inhibitory effect of CompC on PDGF-induced signal transduction and cell proliferation was demonstrated in various PDGFR-expressing cells, including HDFs, MRC-5 human lung fibroblasts, BEAS-2B human bronchial epithelial cells, rat aortic vascular smooth muscle cells and A172 glioblastoma cells [1]. CompC has also been shown to inhibit the viability of human U251 and rat C6 glioma cell lines [17] and glioblastoma cells [4]. Similarly, the present study revealed that CompC, at concentrations of 5–20 μM, reduced the viability of various melanoma cells, such as B16-F1, B16-BL6 and K-1735 murine melanoma cells and G361, DX3, A375P (low metastatic) and A375SM (highly metastatic) human melanoma cells (Figure 1A). B16-F1 is most sensitive to the cytotoxic effect of CompC, whereas normal human epidermal melanocytes NHEMa are less sensitive to that compared to other melanoma cell lines (Figure 1A). Although CompC arrested HDFs at the G0/G1phase of the cell cycle [1], CompC induced cell cycle arrest at G2/M in B16-F1 melanoma cells, as judged by flow cytometry (Figure 1C). Because CompC produced only a minor portion of apoptotic cells in B16-F1 cells (Figure 1C), the present study focused on the mechanism of G2/M cell cycle arrest but not of cell death.

Cell cycle progression is regulated by Cdks and cyclins at each stage (Cdk4/6 and cyclin D1 in G1; Cdk2 and cyclin E/A in S; Cdc2 and cyclin B1 in G2/M) and Cdk inhibitors (p16, p21, p27, and p53). Cdks and cyclins for G1 and S phases are responsible for the hyperphosphorylation of pRb, which promotes cell cycle progression through the G1 and S phases, respectively [18,19]. The phosphorylation of pRb and the subsequent release of the E2F transcription factor complex increases PCNA expression, inducing cell cycle progression through the S phase [20]. Activated Cdc2/cyclin B1 acts in the phosphorylation of histone H3, which plays an important role in the entry of cells into mitosis [21]. Furthermore, Cdk4/cyclin D1 activity can be inhibited by p16, and Cdk2/cyclin E/A activity can be inhibited by p21, p27 and p53 [22]. In addition, binding of p53 to Cdc2 induces downregulation of the Cdc2/cyclins (A or B) protein kinase activities [23,24]. The present study revealed that CompC treatment reduced the levels of most cyclins and Cdks (Figure 2A and Appendix A) and increased the levels of Cdk inhibitors (Figure 2B and Appendix A). The reduced levels of total Cdc2, phosphorylated Cdc2, and increased p21/p53 might be responsible for a reduction of histone H3 phosphorylation (Figure 2A and Appendix A), which finally causes G2/M cell cycle arrest. Because a reduction of upstream Cdks (Cdk4 and Cdk2) and cyclins (D1, E and A) was also observed in CompC-treated cells, we expected a delay in the progression through the G1 and S stages. However, because the level of PCNA was not altered by CompC treatment (Figure 2A and Appendix A), the progression through the S stage in CompC-treated cells could occur without G1 or S arrest (Figure 1C). These results suggested that CompC mainly exerts its inhibitory effect on mitosis via a reduction in Cdc2 activity and subsequent dephosphorylation of histone H3.

Because CompC is an inhibitor of AMPK, the role of AMPK on cell cycle arrest has attracted much research attention. Previously, AMPK activation has been shown to be associated with cell cycle arrest through the activation of p53 and subsequent induction of p21 [3]. However, in our study, CompC, as the AMPK inhibitor, also increased the level of P-p53 at 8-16 h and a delayed increase of p21 at 24 h (Figure 2B and Appendix A). In another experiment with U251 glioma cells, the small interfering RNA (siRNA) targeting of human AMPK was shown to mimic CompC-induced G2/M cell cycle arrest [17]. Conversely, the present study showed that AMPK siRNA treatment alone increased the levels of P-Cdc2 and P-H3 for G2/M cell cycle progression (Figure 2C and Appendix A). CompC-induced reduction of P-Cdc2 and P-H3 was also observed in the AMPK siRNA-treated cells (Figure 2C and Appendix A), indicating that CompC-induced G2/M arrest is AMPK-independent in B16-F1 cells. Similarly, a previous study with AMPK-knockdown MEF cells demonstrated that CompC-induced G0/G1 cell cycle arrest is independent of AMPK inhibition [1]. At present, the effect of CompC on the cell cycle seems to be cell type-specific. This effect remains to be clarified.

Activated Akt plays a role in regulating cell proliferation [25,26] and survival [27,28] by phosphorylating many target proteins [29,30]. Previously, CompC was shown to inhibit Akt in glioma cells [4]. The authors suggested that CompC might increase apoptosis in part by inhibiting the survival signal. Conversely, in our study, CompC increased the level of phosphorylated/activated Akt in B16-F1 cells (Figure 3A and Appendix A). The activation of Akt overcomes cell cycle arrest in G1 [31] and G2 [32] and triggers a network that positively regulates G1/S cell cycle progression through inactivation of GSK3-β, leading to increased cyclin D1, and through inhibition of Forkhead family transcription factors and the tumor suppressor tuberin (TSC2), leading to a reduction of the G1-checkpoint Cdk inhibitor, p27 [25]. Although CompC induced a reduction of Cdks and cyclins involved in G1- and G2-checkpoints (Figure 2A and Appendix A), flow cytometric analysis showed cell cycle arrest at G2/M, but not at G1 and S (Figure 1C). CompC-induced Akt activation might play a role in overcoming cell cycle arrest at the G1 phase. The enhanced Akt activation by CompC might cause an increase in the level of cyclin D1 (Figure 2A and Appendix A) and a decrease in the level of p27 at the early time points, 8-16 h (Figure 2B and Appendix A), which might play roles in overcoming G1/S cell cycle arrest.

Similar to Akt, ERK1/2 are also associated with G1 cell cycle progression. Because CompC increased the phosphorylated/activated ERK1/2 (Figure 3A and Appendix A), it is possible that CompC-induced ERK1/2 might also play a role in overcoming G1 cell cycle arrest. Previously, negative cross-talk between AMPK signaling and ERK signaling has been reported [33]. In addition, cisplatin-induced ERK activation contributes to an elevated level of p53 phosphorylation at Ser^15^ and to p53 accumulation [34]. CompC may activate ERK signaling via inhibition of AMPK, resulting in an increase of phosphorylated p53 (P-p53) in B16-F1 cells (Figure 2B and Appendix A). Furthermore, the involvement of ROS in ERK1/2 activation has been demonstrated [14,35,36]. Therefore, we hypothesized that CompC-induced oxidative stress might be responsible for ERK1/2 activation in an AMPK-independent manner. Indeed, NAC pretreatment reduced CompC-induced P-ERK1/2 levels (Figure 3B and Appendix A) and G2/M-arrested cells (Figure 3C). These data suggest that CompC-induced ROS production (Figure 3D) might play a role in cell cycle arrest at G2/M in B16-F1 cells.

CompC functions as an ATP-competitive inhibitor of AMPK [37], and its selectivity for AMPK is unclear [5]. Moreover, a recent study reported that CompC inhibits a large number of protein kinases, including the ERK8, MNK1, PHK, MELK, DYRK isoforms, HIPK2, Src and Lck, in vitro [38]. Therefore, the inhibitory effect of CompC on other kinases might also be responsible for the CompC-induced reduction of Cdks and cyclins in B16-F1 cells. In addition, a previous study showed that CompC inhibits PDGFR tyrosine kinase [1]. Because there is a structural similarity between PDGFR and VEGFR, having seven immunoglobulin-like extracellular domains [7,15], the effect of CompC on VEGFR signaling in HUVECs may be similar to that on PDGFR signaling in HDFs. The results of the present study clearly demonstrated that CompC inhibits the tyrosine phosphorylation of VEGFR and VEGFR-induced downstream signaling molecules, such as p38, ERK1/2, PLCγ1, PKCθ and PKCμ in HUVECs (Figure 4F and Appendix A), resulting in a reduction of HUVEC viability (Figure 4A). VEGFR signaling is associated with cell proliferation, migration, and capillary tube formation for angiogenesis [6,13]. The inhibitory effects of CompC on HUVEC viability (Figure 4A), tube formation (Figure 5B) and cell migration (Figure 5C–E) and VEGF-induced signaling (Figure 4F and Appendix A) indicate that CompC may have an inhibitory effect on in vivo angiogenesis. As expected, CompC inhibited intratumor vascularization in B16-F1 tumor cell-bearing C57BL/6 syngeneic mice, as judged by light microscopic observation after H&E staining (Figure 5C) and confocal microscopic observation after the immunofluorescence staining of the panendothelial cell antigen MECA32 (Figure 5D). Taken together, the inhibitory effect of CompC on B16-F1 cell growth and angiogenesis might play a role in the reduction of B16-F1 tumor growth in this animal model (Figure 5B).

The receptor tyrosine kinase signaling pathways are major targets in cancer treatment [39]. For instance, sunitinib (also known as SU11248 and marketed as Sutent by Pfizer) is a multitargeted receptor tyrosine kinase inhibitor. Sunitinib inhibits both PDGFR and VEGFR and downstream signaling pathways in tumor cells. Mendel and colleagues reported that sunitinib inhibited in vivo VEGFR phosphorylation in mice bearing A375 melanoma and PDGFR phosphorylation in mice bearing SF767T gliomas [15]. Therefore, as a multitarget oral receptor tyrosine kinase (RTK) inhibitor, sunitinib malate (SU11248) has been used as an anticancer drug, which was approved by the FDA in renal cell carcinoma and imatinib-resistant gastrointestinal stromal tumors [40,41].

Similar to sunitinib, CompC also inhibited both PDGFR signaling in various types of cells [1] and VEGFR signaling in HUVECs, as shown in the present study. Because CompC also blocked B16-F1 tumor cell growth via the inhibition of cell cycle progression, CompC can be used as an anticancer drug targeting malignant tumors, such as melanoma.

## 4. Materials and Methods

### 4.1. Materials

MTT, Minimal Essential Media (MEM) with Eagle’s salts and sodium bicarbonate, mouse anti-β-actin monoclonal antibody (#A5441), PI, RNase A, and 2′,7′-dichlorodihydrofluorescein diacetate (DCFH-DA) were obtained from Sigma-Aldrich (St. Louis, MO, USA). CompC was purchased from Calbiochem (San Diego, CA, USA). Dulbecco’s modified Eagle’s medium (DMEM) and RPMI medium were purchased from Corning (Corning, NY, USA). EGM-2 and EBM were obtained from Clontech (Palo Alto, CA, USA). FBS, antibiotics, M254 melanocyte growth media, HMGS-2 supplements, 1 M Hepes, 100 mM L-glutamine, and 100 mM sodium pyruvate were purchased from Gibco/BRL Life Technologies, Inc. (Carlsbad, CA, USA). Human VEGF-165 and antibodies against Akt (#9272), P-Ser^473^-Akt (#4051), AMPKα1/2 (#2532), ERK1/2 (#9102), P-Thr^202^/Tyr^204^-ERK1/2 (#9106), P-Tyr^783^-PLCγ1 (#2821), Cdk4 (#2906), Cdk2 (#2546), PCNA (#2586), Cdc2 (#9112), P-Tyr^15^-Cdc2 (#9111), P-Thr^161^-Cdc2 (#9114), Cyclin B1 (#4138), p53 (#2524), P-Ser^15^-p53 (#9284), histone H3 (#9715), P-Ser^10^-histone H3 (#3377), VEGFR2 (#9698), P-Tyr^1175^-VEGFR2 (#2478), PKC isotypes θ (#2059) and μ (#2052), P-Thr^538^-PKCθ (#9377), and P-Ser^916^-PKCμ (#2051) were purchased from Cell Signaling Technology (Danvers, MA, USA). Antibodies against cyclins D1 (#sc-56302), E (#sc-481), and A (#sc-751), p53 (#sc-126), p16 (#sc-468), p21 (#sc-397), p38 (#sc-535) and P-Thr^180^/Tyr^182^-p38 (#sc-7973), PLCγ1 (#sc-166938), small interfering RNAs (siRNAs) against mock control and AMPKα1/2, and siRNA transfection reagent were purchased from Santa Cruz Biotechnology (Santa Cruz, CA, USA). Antibodies against pRb (#A00410) and P-Ser^795^-pRb (#A00381) were purchased from Genescript (Piscataway, NJ, USA). Rat antibodies for mouse endothelial antigen MECA32 (#553849) and FITC-conjugated goat anti-rat IgG (#554016), and p27 (#610241) were obtained from BD Pharmingen (Mountain View, CA, USA); mounting solution including DAPI from Invitrogen Life Technologies; Diff-Quik staining solutions from Baxter Diagnostics, Inc. (McGaw Park, IL, USA).

### 4.2. Cell Culture

Mouse B16-F1 (male) and human G361 (male) melanoma cells and primary HUVECs (male) were purchased from American Type Culture Collection (Manassas, VA, USA). B16-BL6 (male) and K-1735 mouse melanoma cells and DX3 (female), A375P (female) and A375SM (female) human melanoma cells were obtained from Korean Cell Bank (Seoul, Korea). NHEMa normal human epidermal melanocytes were supplied by Sun Yeou Kim, College of Pharmacy, Gachon University (Incheon, Korea). B16-F1 and G361 cells were cultured in DMEM and RPMI medium, respectively. B16-BL6, K-1735, DX3, A375P and A375SM cells were cultured in MEM with Eagle’s salts and sodium bicarbonate, which was supplemented with 20 mM Hepes, L-glutamine, and 1 mM sodium pyruvate. DMEM, RPMI, and MEM media were supplemented with 10% FBS and antibiotics. HUVECs were cultured in EGM-2 and NHEMa were cultured in M254 media supplemented with HMGS-2. All cells were maintained in a humidified 5% CO_2_ incubator at 37 °C.

### 4.3. RNA Interference (siRNA)

B16-F1 cells were plated in 6-well plates overnight and the media was replaced with 1 mL of Opti-MEM before transfection. siRNA transfections were performed using the siRNA transfection reagent. Mock control and AMPKα1/2 siRNA duplexes were incubated with 5 μL siRNA transfection reagent for 5 min at room temperature and these mixtures were then added to B16-F1 cells. After incubation for 12 h, 1 mL of Opti-MEM containing 20% FBS was added to each well. At 48 h after transfection with siRNA, the cells were treated with vehicle or CompC for 16 h.

### 4.4. MTT Assay

The cells were seeded in quadruplicate onto 24-well plates at a density of 1 × 10^4^ cells/well, incubated for 1 day and treated with vehicle (DMSO) or various concentrations of CompC for 2 days. Cells were then washed once with PBS and fifty microliters of MTT solution (5 mg/mL stock in PBS) was added to cells, followed by incubation for 1 h at 37 °C in the dark. The medium was removed carefully, and 200 μL of DMSO was added to dissolve the blue formazan. The absorbance was read at 570 nm using an ELISA reader (VICTOR3, PerkinElmer Life and Analytical Sciences, Turku, Finland).

### 4.5. Flow Cytometry

The effect of CompC treatment on cell cycle progression was determined by flow cytometry as described previously [1]. After 1–2 days of culture in 100-mm dishes, cells were treated with vehicle or 10 μM CompC. After incubation for the indicated times, the cells were harvested, washed twice with PBS, fixed in ice-cold 70% ethanol, pelleted by centrifugation, washed with ice-cold PBS, and then incubated with 0.2 mg/mL DNase-free RNase A and 20 μg/mL PI for 30 min at 37 °C in the dark. The cells were analyzed using a FACSCalibur flow cytometer (Becton Dickinson, San Jose, CA, USA) with excitation at 488 nm and emission at 605–617 nm (FL-2 channel).

### 4.6. Western Blot Analysis

The protein levels and phosphorylation status were examined by western blot, as described previously [1]. The cells were treated with FBS or growth factors in the absence or presence of 10 μM CompC, washed twice with ice-cold PBS and lysed in lysis buffer. The lysates (30 μg/lane) were resolved in 8–12% SDS-polyacrylamide gels and transferred to Immobilon PVDF membranes. The blots were analyzed using the appropriate antibodies (1: 1000 dilutions for antibodies purchased from Cell Signaling; 1:200 dilutions for antibodies from Santa Cruz Biotechnology; 1:2000 dilution for monoclonal anti-β-actin antibody) and an ECL detection system.

### 4.7. Measurement of ROS

Cells were incubated with fresh medium containing vehicle or 10 μM CompC for the indicated times. Cells were then administered 25 μM DCFH-DA and incubated for 30 min at 37 °C. After washing with PBS, the cells were trypsinized and washed twice with ice-cold PBS followed by suspension in the same buffer. Fluorescence intensity was measured by flow cytometry using excitation and emission wavelengths of 488 and 525 nm, respectively.

### 4.8. Tube Forming Assay

In 24-well plates, solubilized Matrigel was added to each well and polymerized by incubation at 37 °C for 1 h. The HUVEC cell suspension was added to the Matrigel-coated wells. CompC (1, 5, and 10 μM) was added and incubated for 18 h at 37 °C in a 5% CO_2_ incubator. After tube formation, the cells were stained with Diff-Quik according to the manufacturer’s instructions and photographed with an inverted microscope (Olympus, Tokyo, Japan).

### 4.9. HUVEC Cell Migration

HUVEC cell migration was assessed based on the ability of cells to move into an acellular area, as described previously [42]. HUVECs were seeded onto a 6-cm dish and grown in EGM-2 culture media. After incubation for one day in serum-free EBM to induce quiescence, a few areas were denuded using a sterile yellow tip (1 mm wide), and cells were then incubated in EGM-2 with vehicle or various concentrations of CompC. Photographs of HUVECs were taken at the indicated times, and the number of HUVECs that had migrated to the acellular area was counted using NIH ImageJ software (version 1.43, National Institutes of Health, Bethesda, MD, USA, freely available at http://rsb.info.nih.gov/ij/index.html).

### 4.10. In Vivo Xenograft Experiment

C57BL/6 (age of 5 weeks, male mice) was purchased from Orient Bio Inc. (Gapyung, Korea) and housed in a chamber with controlled temperature and humidity. The animal protocol was reviewed and approved by the Animal Care and Use Committee of Gachon University (GIACUC-R2017005). B16-F1 cells were subcutaneously injected into the shaved right back of C57BL/6 mice. Three days after B16-F1 tumor cell injection, vehicle or CompC was dissolved in isotonic saline, and 100 µL aliquots (corresponding to 2.5 mg/kg/day) were intraperitoneally injected into tumor-bearing C57BL/6 mice anesthetized by methoxyflurane inhalation. The weight and tumor volume of B16-F1 in C57BL/6 mice were examined until 14 days after daily injection, and then the animals were anesthetized with a mixture of Rumpun and Zoletil (1:5). Eight mice per group were sacrificed using a CO_2_ chamber. The tumors were excised and bisected and either fixed overnight in ice-cold 10% paraformaldehyde and embedded in paraffin for hematoxylin–eosin (H&E) staining or frozen in Tissue-Tek^®^ O.C.T. tissue matrix compound (VWR International, Atlanta, GA, USA) for immunofluorescence staining.

### 4.11. H&E Staining

Paraffin-embedded sections were cut, mounted on microslides and processed for H&E, as described previously [42]. The stained slides were dehydrated and mounted using Cytoseal. The slides were photographed (400×) with a microscope (Olympus).

### 4.12. Immunofluorescence Staining and Confocal Microscopy

Frozen tumor samples were cut, mounted on microslides, and air-dried. The slides were postfixed in 100% cold acetone for 2 min. After washing three times with PBS, the slides were treated with 0.3% H_2_O_2_ solution for 10 min, washed three times with PBS and blocked with 1% BSA for 1 h [42]. Tumor sections were incubated with rat antibodies against mouse endothelial cell antigen (MECA-32 clone, 1:200) for 1 h at RT, washed with PBS, and then incubated with FITC-conjugated goat anti-rat secondary antibodies (1:500) for 1 h. Negative control was prepared in the absence of anti-MECA32 antibody. The sections were washed with PBS and dehydrated with 100% ethanol. Coverslips were mounted using Cytoseal mounting solution containing DAPI. The fluorescent images were photographed by laser scanning confocal microscopy (FV500; Olympus).

### 4.13. Statistical Analysis

The Kruskal-Wallis tests for non-parametric statistics (SPSS; IBM, Seoul, Korea) were performed to identify significant difference between vehicle and CompC treatment groups. Post hoc comparisons with Mann-Whitney U tests were used to determine whether the tumor size shows statistically significant difference as a result of the treatment at each day. *p*-values less than 0.05 were considered statistically significant. In vitro data are presented as the mean ± standard deviation of at least three experiments.

## 5. Conclusions

Based on the findings of this study, CompC reduced tumor cell growth by inhibiting the expression and phosphorylation of cell cycle regulatory proteins, such as pRb, Cdks and cyclins, and by increasing the levels of Cdk inhibitors, which resulted in G2/M-phase cell cycle arrest in B16-F1 cells. Moreover, CompC had an inhibitory effect on HUVEC growth, VEGF-induced downstream signaling, and tube formation. Furthermore, the inhibitory effects of CompC on in vivo tumor cell growth and blood vessel formation were confirmed using a C57BL/6 syngeneic mouse model. These results suggest that CompC can be used as an antitumor agent for aggressively growing melanoma.

## Figures and Tables

**Figure 1 cancers-11-00823-f001:**
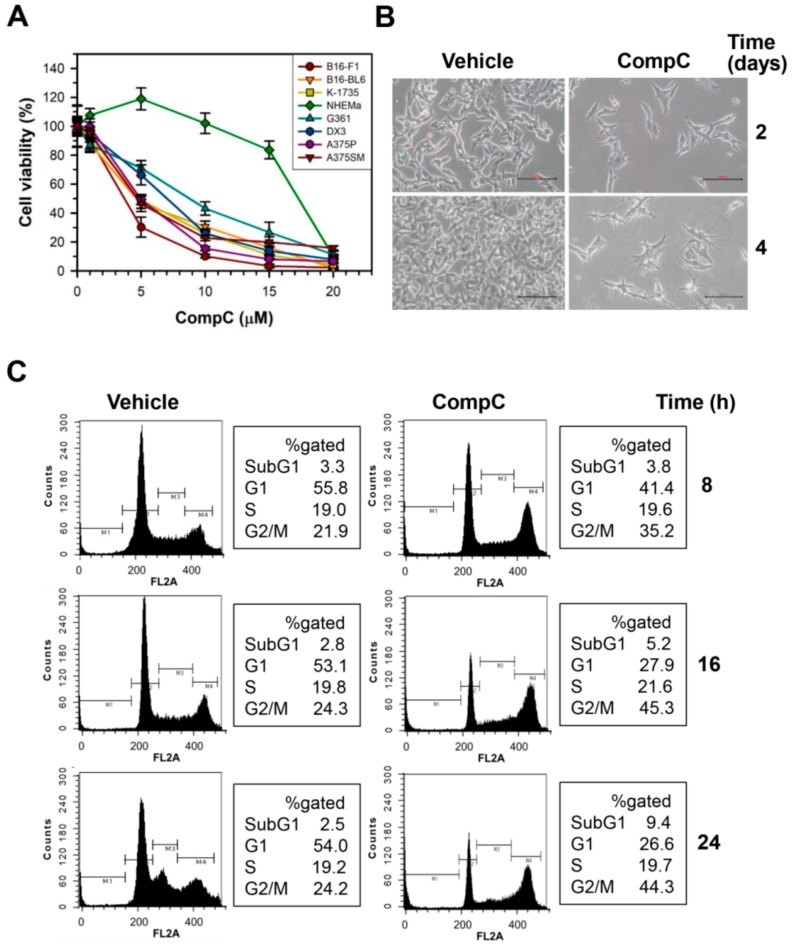
Compound C (CompC)-induced G2/M cell cycle arrest in B16-F1 melanoma cells. (**A**) B16-F1, B16-BL6, K-1735, G361, DX3, A375P, A375SM, and NHEMa cells were cultured for 1 day and treated with vehicle (DMSO) or 1–20 μM CompC for 2 days. An MTT assay was performed and the percentage of cell viability is plotted as the mean ± SD. (**B**) B16-F1 cells were treated with vehicle or 10 μM CompC for the indicated times. The cell morphology was observed by optical microscopy (100×). (**C**) The cells treated with vehicle or 10 μM CompC for the indicated times were stained with propidium iodide, and the DNA content was analyzed by flow cytometry. The percentage of gated area is shown on the right.

**Figure 2 cancers-11-00823-f002:**
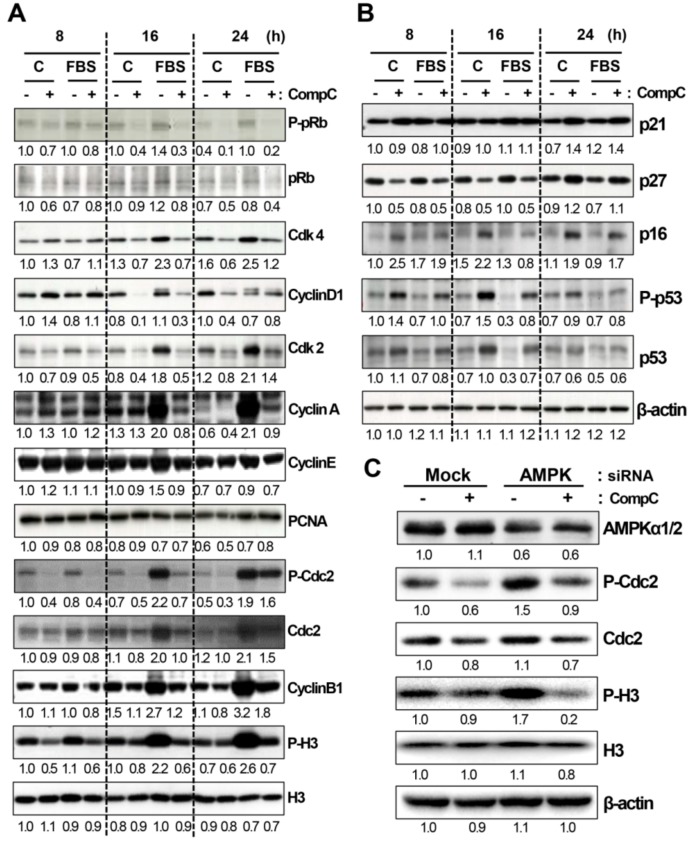
CompC altered the expression and phosphorylation of cell cycle regulatory proteins in an AMP-activated protein kinase (AMPK)-independent manner. (**A**,**B**) Quiescent cells were stimulated with 10% fetal bovine serum (FBS) for the indicated times (8, 16, and 24 h) in the presence of vehicle (−) or 10 μM CompC (+). Cell lysates were analyzed by western blot analysis using antibodies against total and phosphorylated pRb (P-pRb), Cdks, cyclins, and histone H3 (**A**), Cdk inhibitors (**B**), and β-actin as an internal control. The band density of β-actin was used as a normalization control for densitometry analysis of blots in (**A**,**B**). (**C**) B16-F1 cells were transfected with mock or AMPKα1/2 siRNA for 48 h and then stimulated with 10% FBS in the presence of vehicle (−) or 10 μM CompC (+) for 16 h. Cell lysates were analyzed by western blot analysis using antibodies against AMPKα1/2, total and phosphorylated Cdc2 and histone H3.

**Figure 3 cancers-11-00823-f003:**
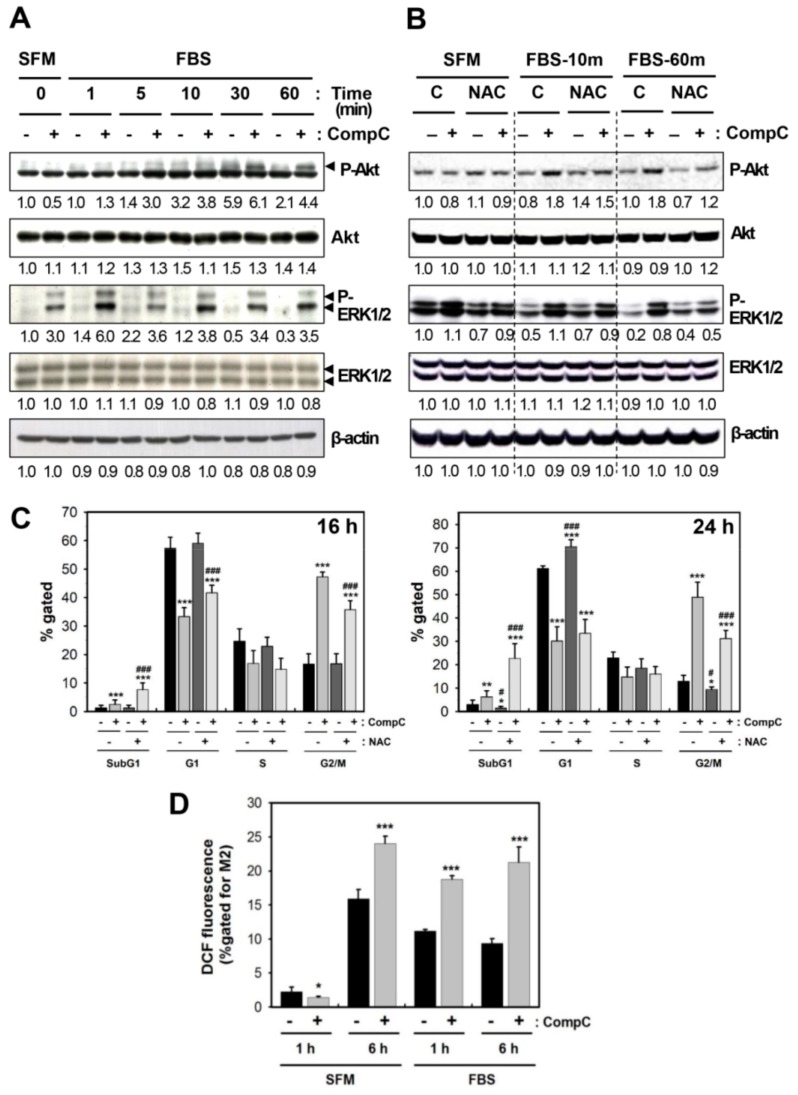
CompC increased the phosphorylations of Akt and ERK1/2 via reactive oxygen species (ROS) production in B16-F1 cells. (**A**) B16-F1 cells were stimulated with 10% FBS for the indicated times in the presence of vehicle or 10 μM CompC. Cell lysates were analyzed by western blot analysis using antibodies against total and phosphorylated Akt (P-Akt) and ERK1/2 (P-ERK1/2). (**B**) Serum-starved cells were pretreated with vehicle alone, 5 mM N-acetyl cysteine (NAC) and/or 10 μM CompC for 1 h. Cells were stimulated with FBS for 10 and 60 min. Cell lysates were analyzed by western blot analysis. (**C**) B16-F1 cells were treated with vehicle or 10 μM CompC in the absence or presence of 5 mM NAC for 16 and 24 h, and the cell cycle was analyzed by flow cytometry. The % gated fractions of sub-G1, G1, S, and G2/M at 16 and 24 h are plotted as the mean ± SD. (**D**) Serum-starved cells were treated with medium containing serum free medium (SFM) or FBS in the presence of vehicle or 10 μM CompC for 1 and 6 h. The level of ROS was measured by flow cytometry and % gated for M2 is plotted. * *p* < 0.05, ** *p* < 0.01, and *** *p* < 0.001, compared with vehicle-treated control cells. ^#^
*p* < 0.05, ^###^
*p* < 0.001, compared with CompC alone-treated cells in the absence of NAC.

**Figure 4 cancers-11-00823-f004:**
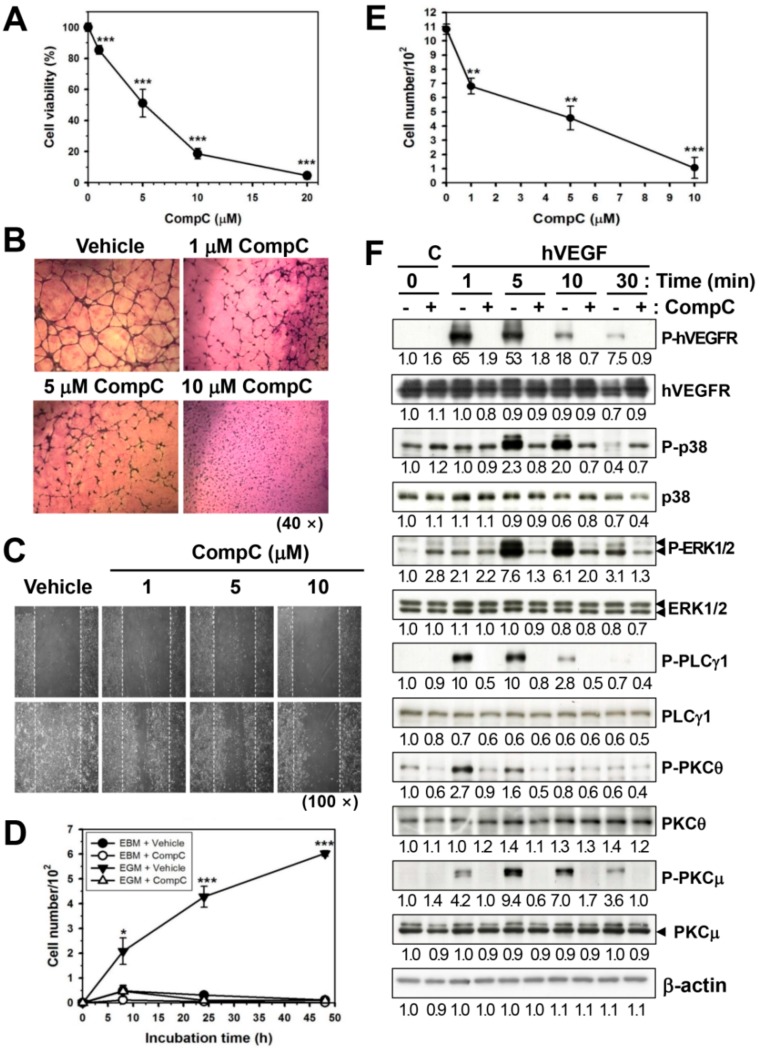
CompC reduced cell viability, tube formation, cell migration, and vascular endothelial growth factors (VEGF)-induced signal transduction in human umbilical vein endothelial cells (HUVECs). (**A**) HUVECs were treated with CompC (1–20 μM), an MTT assay was performed, and the percentage of cell viability is plotted in (**A**). (**B**) The HUVEC suspension was added to Matrigel-coated wells on a 24-well plate. CompC (1–10 μM) was added to endothelial growth media kit 2 (EGM-2) and incubated for 18 h. The cells were stained with Diff-Quik and photographed (40×). (**C**–**E**) HUVECs were grown to 70–80% confluence in a 6-cm culture dish in EGM-2 medium, and some areas were denuded. Cells were then incubated in the culture medium containing various concentrations of CompC and photographed in (**C**) (100×). The number of HUVECs that migrated to the acellular area was counted and plotted in (**D**) (time-dependency) and (**E**) (dose-dependency). (**F**) HUVECs were serum-starved by incubation with endothelial cell basal medium (EBM) for 24 h. Cells were treated with EBM containing 50 ng/mL human VEGF (hVEGF) for the indicated times in the presence of vehicle (−) or 10 μM CompC (+). Cell lysates were analyzed by western blotting with antibodies against total and phosphorylated hVEGF receptor (hVEGFR) and other signaling proteins. * *p* < 0.05, ** *p* < 0.01, and *** *p* < 0.001, compared with vehicle-treated control cells.

**Figure 5 cancers-11-00823-f005:**
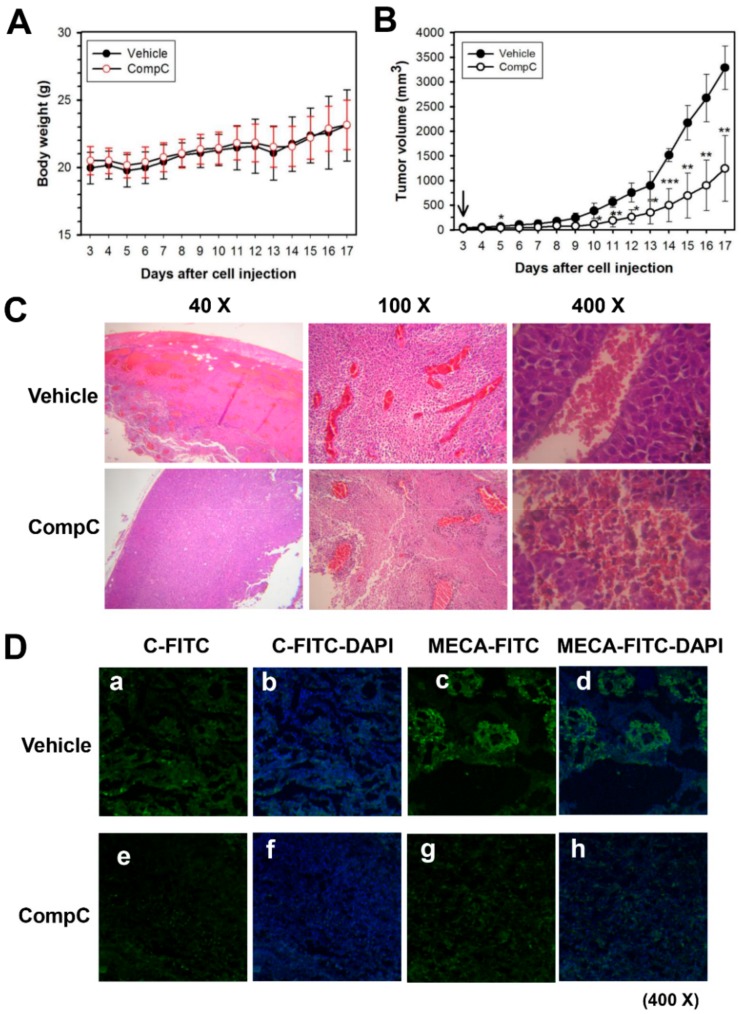
CompC inhibited B16-F1 tumor growth and angiogenesis in C57BL/6 mice. (**A**,**B**) Three days after B16-F1 tumor cell injection, vehicle or CompC (2.5 mg/kg/day) was intraperitoneally injected into the tumor-bearing mice. Body weight and tumor volume were measured and plotted in (**A**,**B**) respectively. Significant difference between vehicle and CompC injection groups at each day is marked with an asterisk (*p* < 0.05, Mann-Whitney U test). (**C**,**D**) On 14 days after daily injection of vehicle or CompC, tumors were excised and either embedded in paraffin for H&E staining (**C**) or frozen in Tissue-Tek O.C.T. tissue matrix compound for immunofluorescence staining (**D**). Tumors stained with H&E (40–100×) or immunostained with (**c**,**d**,**g**,**h**) or without anti-MECA32 antibodies (negative control: **a**, **b**, **e**, and **f**) in the presence of FITC-conjugated secondary antibodies and 4′,6-diamidino-2-phenylindole (DAPI) stain were photographed (400×).

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
