# Peer review of "Compound C Inhibits B16-F1 Tumor Growth in a Syngeneic Mouse Model via the Blockage of Cell Cycle Progression and Angiogenesis"

_cancers, 2019, doi:10.3390/cancers11060823_

Round 1
Reviewer 1 Report
This manuscript examines the effect of CompC on the viability of B16-F1 cells by examining the cell cycle regulatory proteins that are effected after treatment with this compound. The authors' present an extensive amount of data and for the most part the evidence supports their conclusions. Before publication there are a few concerns that should be addressed:
This study would be strengthened if a model for human melanoma were used in conjunction with the murine melanoma cell line. Even if the authors' are unable to perform translational experiments with human melanoma cells, they could at least conduct in vitro experiments using these lines.
It in unclear why 4TI mammary carcinoma cells and Renca murine renal carcinoma cells were included in Figure 1A - these cell lines are not continued in the study - are they meant to serve as a control for CompC treatment?
Figure 1B is of poor quality to see individual cells. The figure looks like a green box with little to no difference between the treated and control groups.
The immunoblots presented in Figure 2 are good; however the blots for H3 appear to be either overloaded or overexposed which could affect densitometry analysis. Also, if densitometry analysis is being done the authors may be able to apply some statistical analysis across replicates to confirm the results are consistent and reproducible.
In the methods section more information should be provided regarding the antibody use and dilutions. Many of these companies sell multiple versions of the same antibody; to insure reproducibility of these results by others more information needs to be included.
The authors are using a murine model but test the effects of CompC on angiogenesis using human endothelial cells. Although one would expect similar results with murine derived endothelial cells, they should be used for consistency between models.
Figure 4C is of poor quality - there is a lot of fluorescence green staining making it difficult to see the scratch and evaluate the migration
Figure 5D lacks negative controls. Negative controls are essential for evaluating the specificity of staining of the primary antibody.
The statistical methods are not appropriate for this data - particularly with respect to measuring tumor volume in vivo. The student T-test relies on parametric data sets to which live cells almost never provide this type of data. It is recommended that the authors consult with a statistician that can help them determine the most appropriate methods to analyze their results.
Author Response
We thank the reviewers for appreciating the significance of the present study and providing constructive suggestions. We respond to the reviewer’s general and specific comments as follows:
Comments and Suggestions for Authors
This manuscript examines the effect of CompC on the viability of B16-F1 cells by examining the cell cycle regulatory proteins that are effected after treatment with this compound. The authors' present an extensive amount of data and for the most part the evidence supports their conclusions. Before publication there are a few concerns that should be addressed:
This study would be strengthened if a model for human melanoma were used in conjunction with the murine melanoma cell line. Even if the authors' are unable to perform translational experiments with human melanoma cells, they could at least conduct in vitro experiments using these lines.
Response: In this study, we used B16-F1 cell line because B16-F1 has been used as a model for the study of most aggressive and high-risk melanoma in many literatures. Furthermore, we were unable to perform an in vivo animal study with human cells. We could carry out further studies with B16-F1 murine melanoma cells and a syngeneic mouse model. As the reviewer 1 suggested, the revised manuscript included new experimental data on other murine and human melanoma cells, including B16-BL6, K-1735, DX3, G361, A375P, A375SM and normal human epidermal melanocytes (NHEMa). As shown in the revised Figure 1A, the MTT assay revealed that CompC significantly reduced cell viabilities of these murine and human melanoma cells and B16-F1 cells were more sensitive to CompC than other melanoma cells. Thus, we have focused on B16-F1 cells for further studies of the effect of CompC on melanoma cells.
It is unclear why 4TI mammary carcinoma cells and Renca murine renal carcinoma cells were included in Figure 1A - these cell lines are not continued in the study - are they meant to serve as a control for CompC treatment?
Response: We had included 4TI mammary carcinoma cells and Renca murine renal carcinoma cells in Figure 1A to show the generalized effect of CompC on cancer cells. As the reviewer 1 suggested, in the revised manuscript, we added the other murine and human melanoma cells and normal human melanocytes instead of 4T1 and Renca cells.
Figure 1B is of poor quality to see individual cells. The figure looks like a green box with little to no difference between the treated and control groups.
Response: We prepared new microphotographs of cells after incubation with 10 μM CompC for 2-4 days. Changes in cell number and cell shape and detachment of cells were clearly visible after treatment with 10 μM CompC for 2 days (Figure 1B).
The immunoblots presented in Figure 2 are good; however the blots for H3 appear to be either overloaded or overexposed which could affect densitometry analysis. Also, if densitometry analysis is being done the authors may be able to apply some statistical analysis across replicates to confirm the results are consistent and reproducible.
Response: As the reviewer 1 mentioned, histone H3 is very abundant protein within the cells so that the blots for H3 appear to be either overloaded or overexposed. Thus, we presented other H3 western blot with better resolution between bands in the revised manuscript. We repeated western blot analysis for H3 several times. Among them, 4 blots are shown below. We performed densitometry analysis using four blots and β-actin band as a control for normalization (shown in Figure 2B), and provided a graph with error bars for H3 western blots as supplementary data (Suppl. Figure S1).
In addition, since we used β-actin as a control for normalization shown in Figure 2B, to clarify the normalization control for blots in Figure 2A, we added a description in the figure legend as follows (underlined sentence):
Figure 2. CompC altered the expression and phosphorylation of cell cycle regulatory proteins in an AMPK-independent manner. (A, B) Quiescent cells were stimulated with 10% FBS for the indicated times (8, 16, and 24 h) in the presence of vehicle (-) or 10 μM CompC (+). Cell lysates were analyzed by western blot analysis using antibodies against total and phosphorylated pRb (P-pRb), Cdks, cyclins, and histone H3 (A), Cdk inhibitors (B), and β-actin as an internal control. The band density of β-actin was used as a normalization control for densitometry analysis of blots in A and B. (C) B16-F1 cells were transfected with mock or AMPKα1/2 siRNA for 48 h and then stimulated with 10% FBS in the presence of vehicle (-) or 10 μM CompC (+) for 16 h. Cell lysates were analyzed by western blot analysis using antibodies against AMPKα1/2, total and phosphorylated Cdc2 and histone H3.
In the methods section more information should be provided regarding the antibody use and dilutions. Many of these companies sell multiple versions of the same antibody; to insure reproducibility of these results by others more information needs to be included.
Response: As the reviewer 1 suggested, we provided more information on the antibody, such as catalog number and dilutions in Materials and methods.
The authors are using a murine model but test the effects of CompC on angiogenesis using human endothelial cells. Although one would expect similar results with murine derived endothelial cells, they should be used for consistency between models.
Response: Because human vascular endothelial cells (HUVECs) have been used as a typical model for angiogenesis and it was difficult to get murine derived endothelial cells, we tested the effect of CompC on angiogenesis in HUVECs. Especially, we have a previous result on the inhibitory effect of CompC on the human platelet-derived growth factor receptor (PDGFR) signaling in human diploid fibroblasts (Kwon et al., Cell. Signal. 2013, 25, 883-897). Because PDGFR and vascular endothelial growth factor receptor (VEGFR) belong to a same receptor family, we postulated that CompC could affect angiogenesis via the inhibition of human VEGFR. To confirm our hypothesis, we used HUVECs which possess a plenty of human VEGFRs. We expect similar results with murine derived endothelial cells.
Figure 4C is of poor quality - there is a lot of fluorescence green staining making it difficult to see the scratch and evaluate the migration
Response: We replaced the green staining with the grey staining. Additionally, we increased the light and contrast within the picture. We hope it is better to see the scratch lines and cell migration.
Figure 5D lacks negative controls. Negative controls are essential for evaluating the specificity of staining of the primary antibody.
Response: As the reviewer 1 suggested, we included negative controls for anti-MECA32-FITC staining (Fig. 5D) and we added this description in the Result section and the figure legend (Figure 5D).
The statistical methods are not appropriate for this data - particularly with respect to measuring tumor volume in vivo. The student T-test relies on parametric data sets to which live cells almost never provide this type of data. It is recommended that the authors consult with a statistician that can help them determine the most appropriate methods to analyze their results.
Response: As the reviewer 1 suggested, the Kruskal-Wallis tests for non-parametric statistics (SPSS; IBM, Seoul, Korea) were performed to identify significant differences between vehicle and CompC treatment groups. Bonferroni-corrected Mann-Whitney U test was used as a post hoc comparison on each date. The statistics was described in the Results, Figure legends, and Materials and methods sections.

Reviewer 2 Report
Lee et al report that the anti-proliferative agent Compound C inhibits cell division of B16-F1 melanoma cells cultured in vitro or xenografted onto C57BL/6 male mice. Furthermore, they find that the drug prevents proliferation, migration and tube formation of human umbilical vascular endothelial cells. The authors provide evidence that cells treated with Compound C accumulate in G2/M and that this partial cell cycle arrest is associated with reduced levels of cell division kinases and cyclins and elevated levels of Cdk inhibitors. This effect may be mediated by Akt and ERK1/2 signaling. Lee et al conclude that Compound C might be a clinically effective agent to treat melanoma.
The title insinuates that Compound C inhibits in vivo melanoma cell growth by deregulating cell cycle regulators. However, only the tumor volume and its vascularisation were assessed in vivo. The cell cycle data were obtained by analysing melanoma cell cultured in vitro where the effect on cell cycle regulators and inhibitors might be different. The data and the conclusions should be presented accordingly.
The Western blot data in Figures 2A-C and 3A,B are quantified but the data are somewhat oddly represented merely by numbers below the images rather than the usual bar diagrams. I suspect that the authors sought to save space when they put the figures together. I get the idea, however, this makes the data much harder to appreciate. I strongly recommend re-organising the figures to represent changing protein levels using a graphical display. The authors should also indicate how often the Western blot experiments were repeated and appropriate error bars should be shown. This also holds true for the assays shown in Figure 4F where no quantification is provided.
Author Response
Responses to the Reviewers’ Comments
We thank the reviewer 2 for appreciating the significance of the present study and providing constructive suggestions. We respond to the reviewer’s general and specific comments as follows:
Comments and Suggestions for Authors
Lee et al report that the anti-proliferative agent Compound C inhibits cell division of B16-F1 melanoma cells cultured in vitro or xenografted onto C57BL/6 male mice. Furthermore, they find that the drug prevents proliferation, migration and tube formation of human umbilical vascular endothelial cells. The authors provide evidence that cells treated with Compound C accumulate in G2/M and that this partial cell cycle arrest is associated with reduced levels of cell division kinases and cyclins and elevated levels of Cdk inhibitors. This effect may be mediated by Akt and ERK1/2 signaling. Lee et al conclude that Compound C might be a clinically effective agent to treat melanoma.
The title insinuates that Compound C inhibits in vivo melanoma cell growth by deregulating cell cycle regulators. However, only the tumor volume and its vascularisation were assessed in vivo. The cell cycle data were obtained by analysing melanoma cell cultured in vitro where the effect on cell cycle regulators and inhibitors might be different. The data and the conclusions should be presented accordingly.
Response: In vitro study demonstrated that both cell cycle progression and angiogenesis are inhibited by CompC treatment. In addition, it is clear that the tumor growth is associated with cell cycle progression. That’s the reason why we added the words: the blockage of cell cycle progression”. Nevertheless, as the reviewer 2 suggested, we changed the title to “Compound C inhibits B16-F1 tumor growth in vitro and in a syngeneic mouse model”.
The Western blot data in Figures 2A-C and 3A, B are quantified but the data are somewhat oddly represented merely by numbers below the images rather than the usual bar diagrams. I suspect that the authors sought to save space when they put the figures together. I get the idea, however, this makes the data much harder to appreciate. I strongly recommend re-organising the figures to represent changing protein levels using a graphical display. The authors should also indicate how often the Western blot experiments were repeated and appropriate error bars should be shown. This also holds true for the assays shown in Figure 4F where no quantification is provided.
Response: Our manuscript has too many blots. If we insert graphical displays into the manuscript, it will seem too complicated. In addition, changes in most major proteins are readily apparent in the blot. That’s the reason we insert the normalized fold numbers only. Nevertheless, as the reviewer 2 suggested, we provided graphs with appropriate error bars as supplementary data.
Reviewer 3 Report
The manuscript entitled "Compound C inhibits B16-F1 tumor growth in a syngeneic mouse model via the blockage of cell cycle progression and angiogenesis" presents in vitro studies of the mechanism of action of CompC on B16-F1 mouse melanoma cells and HUVECs, as well as, in vivo evidence that this compound inhibits indeed tumor growth in xenografts. The studies are well-designed and conducted, and the results are interesting and promising.
I would like to make just two minor comments regarding Figure 3:
1) How do the authors explain the fact that FBS in Fig. 3A does not cause phosphorylation of ERK (comparison of the lanes without CompC in the presence or absence of serum) ?
2) In Fig. 3D what are the effects of NAC on the other cell cycle phases ? The authors should present all cell cycle phases, since it helps to understand the global effect on cell cycle. If they don't want to overload their manuscript, they can add these data as supplementary information.
Author Response
Responses to the Reviewers’ Comments
We thank the reviewer 3 for appreciating the significance of the present study and providing constructive suggestions. We respond to the reviewer’s general and specific comments as follows:
Comments and Suggestions for Authors
The manuscript entitled "Compound C inhibits B16-F1 tumor growth in a syngeneic mouse model via the blockage of cell cycle progression and angiogenesis" presents in vitro studies of the mechanism of action of CompC on B16-F1 mouse melanoma cells and HUVECs, as well as, in vivo evidence that this compound inhibits indeed tumor growth in xenografts. The studies are well-designed and conducted, and the results are interesting and promising.
I would like to make just two minor comments regarding Figure 3:
How do the authors explain the fact that FBS in Fig. 3A does not cause phosphorylation of ERK (comparison of the lanes without CompC in the presence or absence of serum) ?
Response: In fact, it is found that, unlike normal primary cells, the stimulating effects of FBS on ERK phosphorylation were not obvious in many cancer cells, probably due to autocrine action of growth factors in cancer cells. As shown in Figure 3A and 3B, melanoma cells also have basal level of phosphorylated ERK1/2 and FBS treatment slightly increases the level of P-ERK1/2 up to 2.2 fold at 5 min. Thereafter, P-ERK1/2 was reduced to the level lower than the basal level. A certain downregulation system (i.e., phosphatases?) might be activated by FBS treatment. CompC treatment may block the downregulation system. CompC-induced ROS production and ROS-dependent ERK1/2 activation might be a causative factor. We discussed this in the Discussion section as follows:
Similar to Akt, ERK1/2 are also associated with G1 cell cycle progression. Because CompC increased the phosphorylated/activated ERK1/2 (Figure 3A), it is possible that CompC-induced ERK1/2 might also play a role in overcoming G1 cell cycle arrest. Previously, negative cross-talk between AMPK signaling and ERK signaling has been reported [33]. In addition, cisplatin-induced ERK activation contributes to an elevated level of p53 phosphorylation at Ser15 and to p53 accumulation [34]. CompC may activate ERK signaling via inhibition of AMPK, resulting in an increase of phosphorylated p53 (P-p53) in B16-F1 cells (Figure 2B). Furthermore, the involvement of ROS in ERK1/2 activation has been demonstrated [14,35-36]. Therefore, we hypothesized that CompC-induced oxidative stress might be responsible for ERK1/2 activation in an AMPK-independent manner. Indeed, NAC pretreatment reduced CompC-induced P-ERK1/2 levels (Figure 3B) and G2/M-arrested cells (Figure 3C). These data suggest that CompC-induced ROS production (Figure 3D) might play a role in cell cycle arrest at G2/M in B16-F1 cells.
In Fig. 3C what are the effects of NAC on the other cell cycle phases? The authors should present all cell cycle phases, since it helps to understand the global effect on cell cycle. If they don't want to overload their manuscript, they can add these data as supplementary information.
Response: As the reviewer 3 suggested, we included the effects of NAC on the all cell cycle phases (sub-G1, G1, S, and G2/M) at 16 h and 24 h in the revised manuscript (Fig. 3C). We also briefly described the effects of NAC on the other cell cycle phases in the Results section.

Round 2
Reviewer 1 Report
The authors have adequately addressed this reviewer's concerns, the manuscript is now acceptable.